# Gut–Joint Axis: Impact of Bifidobacterial Cell Wall Lipoproteins on Arthritis Development

**DOI:** 10.3390/nu15234861

**Published:** 2023-11-21

**Authors:** Frank Piva, Philippe Gervois, Youness Karrout, Famara Sané, Marie-Bénédicte Romond

**Affiliations:** 1Virology Laboratory-ULR3610, University of Lille and CHU Lille, 59000 Lille, France; frank.piva@univ-lille.fr (F.P.); philippe.gervois@univ-lille.fr (P.G.); famara.sane@chru-lille.fr (F.S.); 2Inserm U1008, University of Lille and CHU Lille, 59000 Lille, France; youness.karrout@univ-lille2.fr

**Keywords:** lipoproteins, *Bifidobacterium longum*, rheumatoid arthritis, microbiome

## Abstract

Gut microbiota affect progression of rheumatoid arthritis (RA). The present study aims at investigating the protective potential of *Bifidobacterium longum* cell wall lipoproteins (Lpps) shown to modulate the intestinal microbiome and prevent osteoarthritis. Arthritis was induced by collagen (CIA) or anti-collagen antibodies (CAIA) injection. Intake of 0.5 mg of Lpps/L, but not 0.25 and 1 mg of Lpps/L, significantly alleviated RA symptoms in CIA DBA/1OOaHsd mice. The arthritis index (AI) was also reduced in CAIA mice. In the CIA-protected group, colon *Ligilactobacillus murinus*, caecal *Lactobacillus johnsonii* and spleen weight correlated with AI, whereas the reverse was observed with splenic CD11c+ dendritic cells (cDCs). The unprotected CIA Lpps group harbored higher cecal and colon *E. coli* and lower caecal *L. murinus*. Lpps administration to CAIA mice after arthritis induction led to lower colon *E. plexicaudatum* counts. Splenocytes from CIA-protected mice triggered by LPS secreted higher Il-10 than control ones. However, a higher IL-10 response was not elicited in gnotobiotic RA mice splenocytes with lower cDCs’ recruitment. Labeled bacteria with the Lpps signal were detected in CIA mice bone marrow (BM) cDCs 5 and 16 h post-gavage but not in Peyer’s patches and the spleen. In vitro uptake of Lpps by primary BM and thymus cells was observed within 24 h. An FACS analysis detected the Lpps signal in the plasmacytoid cell compartment but not in cDCs. In conclusion, Lpps dosing is critical for preventing arthritis progression and appropriately modulating the microbiome. Our results also highlight the possible triggering of the immune system by Lpps.

## 1. Introduction

Autoimmune diseases, particularly type 1 diabetes and rheumatoid arthritis (RA), are associated with altered intestinal microbiota composition [1]. Several recent studies provide evidence that abundance of some taxa increased in the microbiome of RA patients [2,3,4].

Taxa-level alterations can be shared across the inflammatory arthritis phenotypes, e.g., ankylosing spondylitis and psoriatic arthritis [5]. However, there are still contradictory results about the taxa involved. Nevertheless, targeting the microbiota seems promising since the diet, a major factor contributing to the microbial balance, has been shown to affect RA progression and the microbiota composition [6,7,8]. Whereas fasting alleviates the symptoms, food intake is reported to exacerbate disease activity [9]. Supplementation with probiotics can overcome the adverse effect of food intake [10]. 

But screening for efficient probiotics in the context of inflammation is still at the beginning stage. Actually, the arthrogenic potential of intestinal probiotic bacteria explored in the 1980–1990s delayed research on the possible beneficial effect of their oral administration since both lactobacilli and bifidobacteria exhibited pro-arthrogenic properties [11,12,13,14]. New approaches using gnotobiotic animals and a finer analysis of arthrogenic peptidoglycans have led to a better understanding of the underlying mechanisms emphasizing the safety of the oral route [15]. In the late 1990s, *Lactobacilli casei* strain Shirota was demonstrated to prevent arthritis onset [16]. To date, a plethora of studies focusing on lactobacilli showed efficacy in animal models [17,18,19,20,21,22,23,24,25].

In contrast, the protective role of bifidobacteria is less extensively studied. Since RA patients harbor fewer bifidobacteria at the RA onset, their promotion could help prevent disease exacerbation [26]. Administration of *Bifidobacterium pseudocatulatum* prior to the onset of the disease was shown to reduce the arthritis index in a rodent model of arthritis [27]. *B. breve* and *B. longum* intake also alleviate collagen-induced arthritis [28,29].

Still, the respective efficacy of the various strains remains to be ascertained in clinical trials. So far, only the *L. casei* 01 strain has been tested in a clinical trial, showing an improvement in patients’ condition [30]. But, although promising, probiotic safety shall be taken into consideration. There have been reports of several pathological cases recently [31,32,33,34].

The alternative is to employ compounds from bacteria that can emulate the protective effects of the bacteria themselves. The free form of cell wall lipoproteins (Lpps) released by either *Bifidobacterium longum* or *B. breve* during milk fermentation reproduced the effects on the microbiota observed with the whole fermented milk [35]. The released Lpps are complex molecules sharing across the bifidobacterial species a 43–45 kDa protein exhibiting a lipobox and a CHAP domain on the N- and C-terminal sequences, respectively. Di- or tri-acetylation on the lipobox cysteine led to a possible recognition of TLR2 receptors [36]. 

Administration of *B. longum* Lpps free form to gnotobiotic mice associated with the microbiota from an RA patient led to the increase in intestinal bifidobacteria and the restoration of a spleen CD11c+ DC transcriptomic pattern resembling those observed in mice associated with the microbiota from a healthy individual [37]. 

Furthermore, Lpps intake by rats following induction of osteoarthritis protected the animals against the disease’s progression [38]. The results prompted us to further investigate the anti-arthritic potential of the free form of cell wall Lpps using animal models.

The purpose of the present study was to evaluate the Lpps protection against arthritis development and decipher whether (i) gut bacteria related to osteoarthritis development were as well involved, and/or (ii) Lpps were able to reach the inflamed joint area, indicating a possible in situ effect. Free lipoproteins from *B. longum* were administrated in collagen-induced (CIA) and anti-collagen-antibody-induced arthritis (CAIA) DBA/1OOaHsd mouse models and the arthritis index was scored. Selected bacteria were quantified along the intestine at the end of the surveys. To further investigate the possible impact of the microbiome, we included gnotobiotic mice associated with an RA patient’s microbiota as a surrogate of arthritis initiation and we compared IL-10 responses to lipopolysaccharide elicitation of CIA and RA splenocytes. In addition, bone marrow (BM) was targeted as a compartment close to the destroyed joint and possibly infiltrated with an inflammator. BM also comprises dendritic cells (DCs) able to capture Lpps following a possible passage across the intestinal barrier. Uptake of Lpps was studied ex vivo and in vivo by administrating labelled Lpps to CAIA and healthy mice. 

## 2. Materials and Methods

### 2.1. Animals

All experiments were conducted following the 2010 EU directory guidelines and were approved by the Ethical Committee for Animal Experimentation (Lille, France). DBA/1OOaHsd mice were purchased from Harlan France (Gannat, France). C3H/HeJ mice associated with RA donor microbiota (RA mice) were kept in sterile isolators (La Calhène, Vélizy, France) with free access to sterile RO3 pellets (UAR, Epinay-sur-Orge, France) and sterile water in the university animal facility [37]. *B. longum* free lipoproteins were prepared as previously described [38]. Briefly, bifidobacteria were grown in a broth containing lactose at 70 g/L, hydrolyzed milk proteins at 10 g/L and cysteine hydrochloride at 0.03 g/L for 48 h at 37 °C in anaerobic conditions. The supernatant was collected using high-speed centrifugation, concentrated by means of serial ultrafiltration. The retentate comprising molecular weight above 100 kDa was then extensively dialyzed and lyophilized. Residual contamination with broth proteins was checked by subjecting the retentate to gel filtration chromatography on Superdex 200^®^ (Sigma–Aldrich, St. Quentin Fallavier, France).

### 2.2. Arthritis Induction

Arthritis was induced by collagen (CIA) as follows: 5–6-week-old male DBA/1 J mice were intravenously injected with collagen II (MDBioproducts, Zürich, Switzerland) at D0 (with complete Freund’s adjuvant) and D30 [39]. Administration of Lpps started on day 0. Lpps were dissolved in sterile water at 0.25, 0.5 and 1 mg/L, distributed in sterile bottles and kept frozen at −20 °C until use. Bottles were replaced on the cage with new ones daily (*n* = 8 mice per group per assay). The volume left in the bottles removed in the morning was measured for an estimation of the dosing. Body weight was monitored all along the survey. The control group (*n* = 8 per assay) received sterile water instead of the Lpps solution. The mice were monitored daily for the development and severity of joint inflammation (toes, tarsus, ankle, wrist and knee). Each paw was scored on a scale of 0–4 based on signs of swelling and inflammation by 2 observers who were blinded to the group assignments via a visual assessment scoring system (0: no evidence of erythema or swelling, 1: erythema and mild swelling confined to the midfoot or ankle joint, 2: erythema and mild swelling extending from the ankle to the midfoot, 3: erythema and moderate swelling extending from the ankle to the metatarsal joints, 4: erythema and severe swelling encompassing the ankle, foot and digits) [40].

The combined limbs total score was recorded each day (max. score = 16). At the end of the assay (around D45), mice were euthanized and the following organs were collected: kidney, spleen, liver, lung, Peyer patches, ileum, cecum and colon. 

The anti-collagen antibody cocktail was purchased from MD Bioproducts and used according to the manufacturer’s instructions to induce arthritis (anti-collagen antibodies induced arthritis-CAIA model). Male DBA/1OOaHsd mice aged 5–6 weeks were intravenously injected with the cocktail of anti-collagen antibodies. Three days later, mice were intraperitoneally injected with 50 μg of *E. coli* 0111:B4 LPS (Sigma, Saint Quentin-Fallavier, France) to trigger the development of arthritis. Administration of Lpps at a concentration of 0.2 mg/L started 10 days prior to the cocktail injection for the prophylactic Lpps dosing regimen and after the LPS injection (day 0) for the curative Lpps dosing regimen. Control mice received sterile water. Lpps intake was estimated by measuring the remaining volume daily. Mice were monitored and scored daily as described above. At the end of the assay, mice were euthanized and the following organs were collected: kidney, spleen, liver, lung, Peyer patches, ileum, cecum, colon. 

### 2.3. Bacterial Enumeration in Organs 

Organs (intestine fractions, spleen, Peyer’s patches) were weighted and suspended in a 9 mL pre-reduced Ringer solution (Solabia, Pantin, France) supplemented with cysteine HCl (0.03%) (VWR, Fontenay-sous-Bois, France). Except for enterobacteria enumerated onto an EMB medium, bacteria (see Appendix A) were counted using real-time qPCR. The suspensions were kept frozen until total DNA extraction. After thawing, total DNA was extracted using the Nucleospin Tissue kit (Macherey Nagel, Hoerdt, France). DNA content was determined at a 260 nm wavelength using Biophotometer Plus (Eppendorf, Montesson, France). The qPCRs were set out in 25 μL final volumes containing 5 μL of a DNA template (10 ng), 12.5 μL of HOTPol Evagreen qPCRSupermix (Euromedex, Strasbourg, France) and optimized concentrations of the primers [38]. The amplification conditions were 5 min denaturation at 95 °C, followed by 40 cycles of 30 s at 95 °C, 30 s for annealing at a temperature depending on the primer set and 30 s at 72 °C, and a final extension at 72 °C for 7 min (Mastercycler realplex, Eppendorf, Hamburg, Germany). A melting curve analysis and migration onto 2% agarose gel (Eurogentec, Liège, Belgium) of PCR products from samples and isolated control bacteria were carried out. PCR products from isolated bacteria were purified using the Nucleospin Gel and PCR Clean Up Kit (Macherey Nagel) according to the manufacturer’s instructions. NGS was performed using Illumina Miniseq (Genoscreen, Lille, France). The qPCR assays were calibrated using 0.01 fg up to 10 pg (i.e., 42 to 4.2 × 10^7^ copies) of the PCR-amplified target gene sequence from corresponding taxa.

### 2.4. Enumeration of Spleen CD11C+ DCs and CD4+ T Cells and Culture of Splenocytes in CIA Model

Primary splenocytes were aseptically isolated from CIA mice (control group, *n* = 10; 0.5 mg Lpps/L group, *n* = 11; 1 mg Lpps/L, *n* = 6) or from RA mice treated (*n* = 7) or not (*n* = 9) by Lpps. Briefly, the spleen was dilacerated in sterile PBS and digested with 2 mg/mL of collagenase (Sigma, France). CD11c and CD4 (L3T4) cells were positively separated using MACS^®^MicroBead Technology (Miltenyi Biotec, Paris, France). CD4+ CD25+ regulatory T cells were further positively separated from RA cell suspensions. In addition, splenocytes from four control and five CIA mice receiving 0.5 mg of Lpps/L as well as four control and four Lpps-treated RA male mice were kept in RPMI 1640 broth (Gibco^®^, Villebon-sur-Yvette, France) supplemented with fetal bovine serum (5%) (Gibco^®^) and 50U penicillin–streptomycin (Gibco^®^). Suitable cell numbers were evaluated using the trypan blue exclusion method and a hemocytometer. PBS-washed cells were exposed to the fresh medium supplemented with or without lipopolysaccharides from *E. coli* O111:B4 (Sigma) (prepared in pH 7.2 PBS) ranging from 2 ng/mL up to 10 µg/mL. For the measurement of the IL-10 level in the cell culture medium, approximately 2 × 10^5^ cells/well of splenocytes were cultured in a 96-well plate. The cytokine was quantified with an ELISA kit (R&D Systems, Minneapolis, MN, USA) using the manufacturer’s protocol.

### 2.5. In Vivo and In Vitro Lpps Uptake 

Lpps were labelled according to the manufacturer’s instructions with either tetramethylrhodamine-5(6)- isothiocyanate (TRITC) (Enzo Life Sciences, Lyon, France) or Pacific blue (Life Technologies, Carlsbad, CA, USA). The conjugates were stored in the dark at −20 °C until use. 

CAIA DBA/1OOaHsd mice received with gavage TRITC-labeled Lpps at doses ranging from 35 to 100 μg/kg, or sterile water (control mice) supplemented with 10 µL of cell-permeable fluorescent dye, 5′-carboxyfluorescein succinimidyl ester (CFSE). Mice were euthanized 3 h, 5 h or 16 h post-gavage. Peyer’s patches, spleen and bone marrow (BM) were suspended in an RPMI medium and the suspensions were sieved in 70 µm and 40 µm devices. CD11c+ cells were isolated from splenocytes and BM cell suspensions using MACS^®^MicroBead Technology (Miltenyi, France). Cell numbers were evaluated using the trypan blue exclusion method and a hemocytometer. Bacteria and Lpps-containing cells were then evaluated using a fluorescence microscope (Nikon Eclipse E600, Nikon Europe B.V., Amstelveen, The Netherlands). In addition, primary cells from the bone marrow and thymus were aseptically isolated from CAIA mice. Suitable cell numbers were evaluated using the trypan blue exclusion method and a hemocytometer. PBS-washed cells were exposed to a fresh RPMI 1640 medium supplemented with or without Pacific-blue-labeled Lpps (0.15 and 0.3 mg/L). 

A second in vivo analysis was performed in healthy and CAIA DBA/1OOaHsd mice fed R210 pellets (UAR) to avoid contaminating food fluorescence. They were gavaged at doses ranging from 25 to 50 µg/kg with Pacific-blue-labeled Lpps or a broth extract and euthanized 15–17 h post-gavage. CD11c- and mPDCA-positive cells were isolated using MACS^®^MicroBead Technology.

To confirm the possible uptake in steady conditions, healthy mice were gavaged with Pacific-blue-labeled Lpps or a broth extract and euthanized 15 or 72 h post-gavage. The cell suspension was analyzed using a FACSAria™ cell sorter (filter 450/40) (BD biosciences, Le Pont de Claix, France) with the following antibodies: anti-CD4 conjugated to Alexa fluor 647, anti-CD8a conjugated to APC/Cy7, anti-CD11c conjugated to Brillant Violet, anti-PDC-TREM conjugated to phycoerythrin (PE) (BioLegend, San Diego, CA, USA). 

### 2.6. Statistical Analysis

The data were analyzed using SAS Studio. Differences in arthritis index, body weight gain and splenocytes’ response to Lpps were analyzed using a mixed model for repeated measures. Clustering of the data (body weight, spleen weight, splenocytes/mg spleen, bacteria in any intestinal location, splenic CD11c+ and CD4+ cell counts, number of Peyer’s patches) was analyzed using a principal component analysis. Correlations were assayed for their statistical significance using Pearson (spleen weight, CD11c+ DCs, CD4+ T, CD4+ CD25+ cells), Spearman and Kendal’s tau test (bacterial data). The difference in microbial and cell counts was analyzed with the Kruskal–Wallis test or nonparametric analysis of variance followed by the adapted post hoc tests (Bonferroni or Wilcoxon and Kolmogorov–Smirnov). *p* values less than 0.05 were considered significant. 

## 3. Results

### 3.1. Protection against Arthritis Progression Depends on the Lpps Dose and Is Primarily Related to Ligilactobacillus murinus Intestinal Colonization

The statistical analysis of the AI curves using the mixed model for repeated measures (Figure 1A) indicated that Lpps dosing at 0.5 but not 1 mg/L alleviated CIA progression (*p* < 0.05). No protection was observed using 0.25 mg of Lpps/L. An increase in body weight was significant for the three treatments (*p* < 0.0001, Figure 1B). Nonetheless, the body weight curves from both groups drinking Lpps showed a significant higher increase as compared with the control one (*p* < 0.0001). The principal component analysis (PCA) of data including bacterial counts, body weight and spleen characteristics (weight, splenocytes, CD11C+ dendritic cells (DCs) and CD4+ T cells) did not show clustering according to the treatment (Figure 1C). But PCA depicted different bacterial or cell contribution to AI in the three groups. In the CIA-protected 0.5 mg/L Lpps group, colon *Ligilactobacillus murinus* (*p* = 0.0109), cecal *Lactobacillus johnsonii* counts (*p* = 0.0299) and spleen weight (*p* = 0.0285) correlated with AI. On the other hand, a negative correlation between spleen CD11c+ DCs and AI was noted (*p* = 0.0347). *L. murinus* counts in the cecum also negatively correlated with AI (*p* = 0.017). Of note, spleen CD11c+ DCs negatively correlated with colon *L. murinus* (*p* = 0.0188). In the unprotected 1 mg Lpps/L treated group, ileal *E. coli* counts as well as the number of bacteria contaminating the Peyer’s patches negatively correlated with AI (*p* = 0.0114 and *p* = 0.0075, respectively). Interestingly, the control group did not exhibit significant correlation with bacterial counts. However, in contrast with the positive correlation observed in the responsive treated group, spleen weight was negatively correlated with AI (*p* = 0.0448). Only a few differences in the bacterial counts colonizing the intestine were observed (Figure 1D, Appendix A). Actually, variations were primarily detected in the 1 mg Lpps/L group with an increase in cecal *E. coli* (*p* = 0.0397) and a decrease in cecal *L. murinus* (*p* = 0.0026). 

### 3.2. The Degree of Disease Progression Was Significantly Milder in Both Preventive and Therapeutic Lpps CAIA Groups

Protection induced by 0.5 mg/L of Lpps was similar in the preventive and curative groups (Figure 2A). Clustering of the data according to the treatment was not observed (Figure 2B). In addition, the contribution of the microbiome to the arthritis progression needs to be further explored, the set of selected bacteria showing no significant correlation with AI. The sole correlation noted in the preventive Lpps-treated group was between AI and the body weight (*p* = 0.0335). A significant difference in *E. plexicaudatum* counts in the colon was however observed (Appendix A). No *E. plexicaudatum* was detected in the curative group (*p* = 0.0325). 

### 3.3. Higher IL-10 Splenocyte Response to LPS in CIA Lpps-Treated Mice Was Likely Related to CD11c+ DC Recruitment

Negative correlation between CD11c+ DC recruitment and AI prompted us to further analyze the splenic response to Lpps intake. Splenocytes were collected in CIA control and Lpps-treated mice as well as in control and Lpps-treated gnotobiotic mice associated with the microbiome of a patient with rheumatoid arthritis (RA mice) kept in isolators of the animal facilities. Splenocytes were triggered by increasing lipopolysaccharides (LPSs) in the culture broth (Figure 3A). All splenocyte groups showed a significant increased IL-10 response to the LPS augmentation (*p* < 0.0001 each). Moreover, Lpps administration to mice was shown to elicit a higher IL-10 response in CIA splenocytes (*p* = 0.0209), but not in RA splenocytes. 

CD11c+ and CD4+ cells were enumerated in splenocytes from Lpps-treated or control CIA and RA male mice. CD11c+ DCs were significantly lower in RA splenocytes compared with CIA splenocytes (Figure 3B; *p* = 0.0017). Interestingly, gnotobiotic mice associated with a human microbiome from a healthy volunteer were shown to harbor in their spleen CD11C+ DC counts similar to DBA mice, i.e., 4.76% (SD 1.6). It indicated that recruitment of conventional DCs depends primarily on the microbiome.

In contrast, CD4+ T cell compartments were similar in both murine lines regardless of the treatment. Nevertheless, Lpps administration did not affect the recruitment of CD11c+ DCs in the CIA and RA spleen. Conversely, CD11c+ DCs correlated with CD4+ T cells in the CIA (r = 0.9, *p* = 0.0133) but not the RA control group. In 0.5 mg Lpps/L CIA treated mice, CD11c+ DCs still correlated with CD4+ T cells (r = 0.82, *p* = 0.0449) but not in 1 mg Lpps/L CIA treated mice. 

Since RA mice received for a short time the Lpps solution, duration of intake could be too short for detecting an effect on cell recruitment in the spleen. Female mice usually responded more rapidly to microbiome changes. Therefore, in addition to male RA mice, CD4+ T cells and CD4+ CD25+ regulatory T cells were analyzed in splenocytes from RA female control (*n* = 4) and 1 mg of Lpps/L treated (*n* = 3) mice (Figure 3C). In control RA mice, CD4+ CD25+ Treg cells showed higher recruitment in female than male spleens (*p* = 0.0164). When both groups received 1 mg of Lpps/L for 15 days, the female mice showed a lower recruitment of CD11c+ DCs (*p* = 0.029) and CD4+ T cells (*p* = 0.0315) as compared to their male counterparts. Furthermore, female but not male mice responded to Lpps intake by reducing the recruitment of CD4+ T and CD4+ CD25+ Treg cells. Nonetheless, CD11c+ DCs correlated with CD4+ T cells (*p* = 0.0269) and CD4+ CD25 Treg (*p* = 0.0263) in Lpps-treated mice but not control ones. In contrast, CD4+ T and CD4+ CD25+ Treg correlated in the control (*p* = 0.0429) but not in the Lpps group.

To sum up, the difference in IL-10 response to LPS is likely related to the poorer recruitment of CD11C+ DCs in the RA spleen as compared with the CIA one. Moreover, female RA mice responded to Lpps by significantly reducing T cells, more specifically, Treg cell recruitment, both cell compartments depending on CD11c+ Dc recruitment as suggested by their positive correlations.

### 3.4. Lpps Was Primarily Detected in Bone Marrow Plasmacytoid DCs at a Late Post-Gavage Stage

CAIA mice were force fed with a mixture of TRITC-labeled Lpps and CFSE at the end of the week following the LPS booster. Peyer’s patches, spleen and bone marrow were collected 3, 5 and 16 h post-gavage. CD11c+ DCs were separated from splenocytes and bone marrow cells. In addition, two mice receiving only TRITC-labeled Lpps and sacrificed 16 h post-gavage were used for monitoring the possible uptake of fluorescent food components. No significant difference was observed in the DC counts regardless of the organ (Figure 4A). However, DC counts in bone marrow correlated with the arthritis index (*p* = 0.0161). A red signal was detected only in bone marrow 5 and 16 h post-gavage (Figure 4B). More particularly, 16 h post-gavage, the mouse with low AI did not exhibit a detectable red signal. It is noticeable that TRITC fluorescence was always detected in CD11c+ DCs exhibiting a green fluorescence that indicated a parallel capture of bacteria and Lpps. Notably, mice force-fed with TRITC-labeled Lpps deprived in CFSE showed lower DC content. The labeled Lpps were not detected in DCs.

The in vitro assay using cells from the bone marrow and thymus from arthritic CAIA showed a rare uptake by cells from the thymus and bone marrow (0.15 µg/L after 24 h) (Figure 4C). Dying cells were identified by the production of green and red fluorescence. Pacific blue fluorescence was observed in living cells not emitting green or red autofluorescence.

A complementary in vivo assay was performed in CAIA and naïve mice, focusing on conventional CD11c+ DCs and plasmacytoid mPDCA cells from bone marrow, to work out which immune cell is able to capture Lpps. A first separation using MACS^®^MicroBead Technology (Miltenyi Biotec, Paris, France) showed that the distribution of DCs was not similar in the two groups. CD11c+ DCs expanded in bone marrow of CAIA mice as compared with naïve ones (Figure 4D). Plasmacytoid cells were, on the contrary, at the same percentage in the two groups. A first tracking of cells harboring blue fluorescence showed that gavage with the labeled medium’s extract led to the detection of the signal in the sole CD11c+ DCs. Gavage with Pacific-blue-labeled Lpps demonstrated an imbalance distribution in the two DC subpopulations with 6.9% mPDCA in CAIA BM harboring the signal vs. 1.4% CD11c+ DCs, and 2.1% vs. 1.9% in naïve BM, respectively. 

Since contamination from the culture medium cannot be avoided, cells with a Pacific blue signal from naïve mice BM were sorted using FACS. We confirmed that CD11c+ DCs did not exhibit a blue fluorescent signal after gavage with a dose of 25, 50 and 100 µg of Lpps/kg at 15 and 72 h post-gavage. The signal was however detected in the BM cell population as compared with the signal observed in BM cells from control mice force-fed with water (Figure 5A,B). Actually, sorting the cells according to their size allowed for a better Pacific blue detection 15 h post-gavage in the compartment comprising the largest cells (14.5% and 13.4% of the total BM cell suspension, e.g., less than 20% of the cell population). The latter constituted around 96–98% of PDC TREM (Figure 5C). Thus, uptake of Lpps was not related to conventional DCs. Plasmacytoid cells were more likely involved in the capture.

## 4. Discussion

As a whole, our results provided evidence that free lipoproteins released from the cell wall of *B. longum* during fermentation alleviated the course of arthritis in acute (CAIA) or chronic (CIA) phases. But contrary to the mechanism observed in the osteoarthritis model, involvement of the selected gut bacteria already shown to have a prominent role following Lpps intake was limited in animals with an arthritis background [38]. The protective dosage’s initial stages may not be related to intestinal bacteria, but rather to the ability to transport molecules to the inflammatory site, which is filled with antigen-presenting cells like CD11c DCs as shown by their higher number in BM as compared with naïve mice. It certainly raises the question of the transfer of our results to humans.

The mean rate of Lpps release is around 10 mg/11 log_10_ cfu *B. longum* in fermented milk (unpublished personal data). Estimation in a dairy product is around 10 µg per ml. Daily intake of the average content of a yogurt (100–200 mL) could therefore deliver around 15–30 µg/kg per day (adult body weight around 70 kg). The range of the efficient dose in mice was comprised between 50 and 100 µg/kg. Actually, increasing the Lpps supplementation above 1 mg/L aggravated the AI in CIA mice. Therefore, to design appropriate clinical trials, especially the range of the efficient dose, a key point is to delineate the sources of Lpps. Since no bifidobacteria were detected in the intestine of the two arthritis models, the drinking solution was the unique source of Lpps. In contrast, even if the bifidobacterial population is reduced in patients with rheumatoid arthritis (RA), they are still detectable [26]. Therefore, in vivo release of Lpps by indigenous bifidobacteria needs to be confirmed for adjusting the Lpps dosage. 

Exploiting the capacity of Lpps to neutralize virus infection by binding to the viral capside, we developed an affinity column coupled with a virus suitable to capture Lpps in various human secretions [41]. So, we were able to detect free forms of Lpps in human feces either from *B. breve* or *B. longum* (preliminary results). As a consequence, developing a functional food supplement for human RA patients will require determining the intestinal free Lpps content; otherwise, activation of the illness is to be expected as a consequence of a too high Lpps dosage. A first approach relies on the quantification of intestinal bifidobacteria. Still, determination of the bifidobacterial carriage is not predictive of the free Lpps content. On one hand, all bifidobacteria do not display in their genome a homologous protein sequence with a lipobox [42]. Accordingly, the even highly concentrated cell-free whey from *B. bifidum* species deprived in the homologous sequence with a lipobox was unable to modulate in vivo the microbiota [43]. Hitherto, focusing on the secreting bifidobacterial species will not provide an accurate estimation, as the factors tuning in vivo cell wall Lpps release remain to be identified. Actually, the cluster of genes surrounding the sequence coding for the Lpps protein comprises genes involved in the peptidoglycan synthesis and ABC transporters, suggesting that the highest rate of Lpps protein translation occurs during the cell division phase [44]. But factors modulating the post-translational events (such as linkage of the protein sequence to O-oligosaccharides and fatty acid chains) are poorly known, not to mention those involved in the release of the mature Lpps (likely, in vivo, related to some extent to starvation and/or competition for substrates). Nonetheless, regarding prior envisaging to improve in situ Lpps release, further studies are needed. In the meantime, a sensible approach remains as the food supplementation. 

A second aspect is the narrow range of the efficient dose. Two main issues need to be envisaged: the possible too high uptake by BM immune cells reinforcing the inflammation and the susceptibility to oxidative conditions [45]. The inflammation related to the arthritis induction is characterized by the release of oxidate compounds, with possible deleterious effects on the ingested Lpps. Like probiotics, Lpps could be protected through micro-encapsulation, which creates a favorable local environment that enables the encapsulated products to remain functional until they reach their site of action [46]. It can also prove to be useful for delaying Lpps uptake in the BM.

As a matter of fact, our results highlighted the subsidiary impact of the microbiome compared to the role of Lpps in alleviating the arthritis symptoms. First, the bacteria selected for their capacity to affect the progression of osteoarthritis did not contribute to the same extent to the alleviation of arthritis in both acute (CAIA) and chronic (CIA) models [38]. In the acute model, reduced *E. plexicaudatum* counts in the colon of the sole mice receiving Lpps after the arthritis induction, although both groups were protected, indicated that transient change in the microbiome can occur without impacting the illness. In the chronic model, similar findings were gathered, drastic bacterial changes being observed primarily in the unresponsive treated group. However, as reported above, the microbiome constituted a key factor controlling conventional DC recruitment in the spleen and affecting the anti-inflammatory IL-10 response to lipopolysaccharides in the responsive CIA group. Furthermore, the highest spleen DC counts contributed to the lowest arthritis score. Interestingly, PCA illustrated the influence of colon *L. murinus* in reducing spleen DC recruitment and at the same time favoring arthritis progression. 

Intestinal bacteria were also transporters of Lpps as suggested by detection of two fluorescent signals in BM DCs following force feeding with CFSE and labeled Lpps. Surprisingly enough, spleen conventional DCs did not exhibit the two signals, indicating that bacteria carrying Lpps were directed to the BM location close to the damaged joint. Results from the gnotobiotic model associated with the RA microbiome and CIA force-fed mice already suggested that the spleen participated to a lesser degree than bone marrow in the regulation of the inflammation. The female response indicated a reduction in Treg cells in the spleen following Lpps treatment likely related to the changes in gene expression of the splenic cDCS [37]. 

However, the bacterial uptake within BM conventional DCs questioned the degree of cell maturity. Conventional DCs (cDCs), which include the cDC1 and cDC2 subsets, derive from pre-committed BM precursors. The pre-cDC seed lymphoid and non-lymphoid tissues where they further differentiate into mature cDC1 and cDC2. Processing and presenting antigens in the class II pathway is dedicated to the thymus and spleen following Toll-like receptor (TLR) stimulation [47]. As a matter of fact, in RA patients, the synovial inflammatory tissue can reach the adjacent bone marrow by fully breaking the cortical barrier, which results in formation of B-cell-rich aggregates. Plasma cells are present in the regions between aggregates and inflammatory tissue [48]. The bacterial capture in BM from arthritic mice can therefore depend on the inflammatory tissue. 

Nonetheless, a bacterial carrier was not a pre-requisite for transporting Lpps, since BM and thymus cells were shown to capture Lpps added alone to the culture medium. The FACS analysis also emphasized the absence of the Lpps signal in BM CD11c+ DCs, indicating that the cells involved in Lpps uptake belonged to another compartment, almost certainly the plasmacytoid cells. It is already known that PDC-deficient mice show exacerbation of inflammatory and arthritis symptoms [49]. Conversely, enhancing PDC recruitment and activation to arthritic joints by topical application of the Toll-like receptor 7 (TLR-7) agonist imiquimod significantly ameliorated arthritis. The oral administration of Lpps, their detection in the BM PDC TREM compartment and their absence in Peyer’s patches’ cells suggested that a route involving Lpps passage across the intestinal cell layers before reaching BM through the blood stream could prevail. Since pDCs complete their differentiation within the BM, it is most likely that they took up the Lpps as a soluble antigen [50,51]. 

To sum up, our results highlighted the involvement of pDCs as a major contributor to Lpps-induced protection, with some gut bacteria modulating the response via their action on splenic conventional DCs.

## 5. Conclusions

Although promising, this work requires confirmation in humans, particularly through clinical trials, to determine the effective dose under conditions that are compatible with dietary practices. The key point that remains to be clarified is the presence of lipoproteins in the gut. This depends not only on bifidobacterial species colonizing the intestine of healthy individuals or patients with rheumatoid arthritis but also on unknown factors regulating the lipoprotein release at the bacterial level. Furthermore, a more thorough comprehension of the role of plasmacytoid cells in Lpps-induced protection is necessary for future applications in the field of functional foods.

## Figures and Tables

**Figure 1 nutrients-15-04861-f001:**
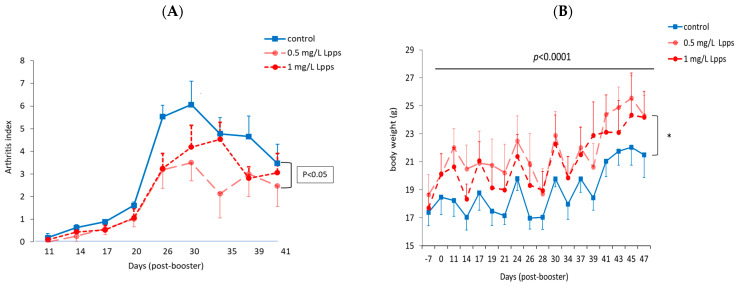
Dosing of 0.5 mg of Lpps/L prevented the progression of arthritis in CIA mice without significant involvement of intestinal bacteria. (**A**) Arthritis index curve of 0.5 mg Lpps/L group is significantly different from that of the control curve (vertical bracket, *p* < 0.05, mixed model for repeated measures). (**B**) the body weight increased significantly in all groups (line above the curves, *p* < 0.0001); however, the weight gain was greater in the two Lpps-treated groups (vertical bracket, *: *p* < 0.0001) as compared with the control group. (**C**) Principal component analysis showed different patterns according to the treatment. Factors (bacteria, cell counts …) contributing significantly to the arthritis score are indicated by a color (red: significant positive correlation, blue; negative correlation). Factors unrelated to AI are labeled by the following abbreviations: BW: body weight; SW: spleen weight; S/S: splenocytes/mg spleen; E1: *E. coli* distal ileum; E2: *E. coli* cecum; E3: *E. coli* colon; D1: *B. fragilis* distal ileum; D2: *B. fragilis* cecum; D3: *B. fragilis* colon; R1: *Ruminococcus* sp. distal ileum; R2: *Ruminococcus* sp. cecum; R3: *Ruminococcus* sp. colon; G1: *L. murinus* distal ileum; G2: *L. murinus* cecum; G3: *L. murinus* colon; H1: *L. reuteri* distal ileum; H2: *L. reuteri* cecum; H3: *L. reuteri* colon; F1: *L. johnsonii* distal ileum; F2: *L. johnsonii* cecum; F3: *L. johnsonii* colon; PP: Peyer’s patch microbiota. (**D**) Bacteria showing difference in counts according to the treatment are illustrated by box plots representing mean (SD) of bacterial counts from 8 mice per treated and control (water) group. The unprotected 1.0 mg Lpps/L treated mice showed increased colonization with *E. coli* in cecum as compared to control group (*p* = 0.0397) and a drastic drop of *L. murinus* in cecum (*p* = 0.0026) as compared to control and 0.5 mg Lpps/L groups).

**Figure 2 nutrients-15-04861-f002:**
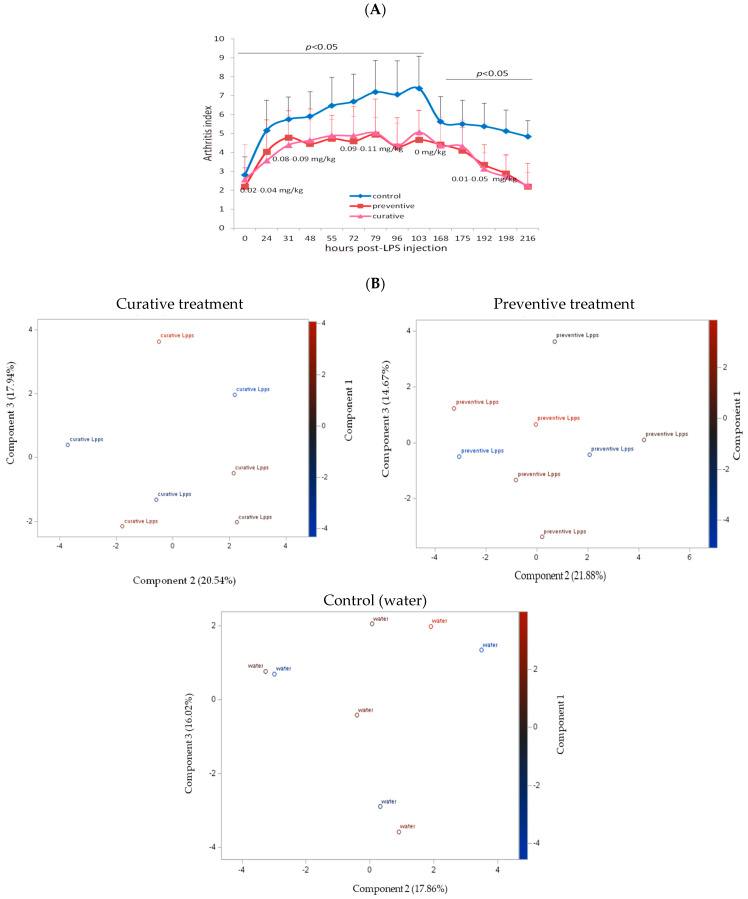
Administration of 0.2 mg/L of Lpps prior to or after LPS injection protected CAIA mice against the progression of acute arthritis without significant involvement of the intestinal bacteria. (**A**) Estimation of the dosing reported in the graph was made by, on a daily basis, weighing the animals and monitoring the volume of Lpps solution consumed. (**B**) The PCA graphs illustrate the localization of animals as featured by the data according to three components.

**Figure 3 nutrients-15-04861-f003:**
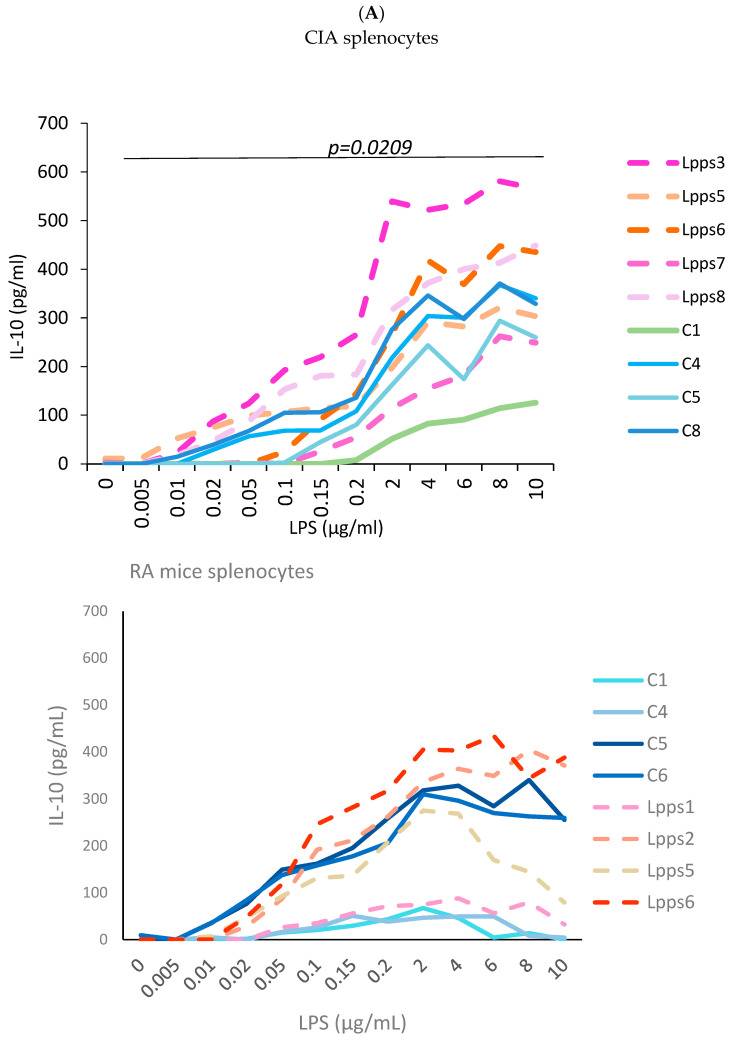
Lpps improvement of the response to LPS in CIA splenocytes was likely related to the DC compartment. (**A**) Splenocytes were collected from 4 control (C1, C4, C5 and C8) and 5 Lpps-treated (Lpps3, Lpps5, Lpps6, Lpps7, Lpps8) CIA mice and from 4 control (C1, C4, C5, C6) and 4 Lpps-treated (Lpps1, Lpps2, Lpps5, Lpps6) RA mice. Increased concentrations of LPS elicited increased IL-10 released in splenocytes from CIA and RA mice (*p* < 0.0001 each). lL-10 response is stronger in CIA mice treated with Lpps (*p* = 0.0209). (**B**) There is a group effect, i.e., splenocytes from RA mice exhibited fewer CD11c+ DCs than CIA mice (*p* = 0.0017). The dosing did not significantly affect CD11c+ DC counts. (**C**) Female RA mice responded to Lpps intake by reducing CD4+ T and CD4CD25 Treg cells (*p* = 0.0273 and 0.0268, respectively). Their spleen contained a greater number of CD4CD25+ Treg than male ones (*p* = 0.0164) prior to treatment. In contrast, following treatment, the female spleen contained less CD11c+ DCs than male ones (*p* = 0.029).

**Figure 4 nutrients-15-04861-f004:**
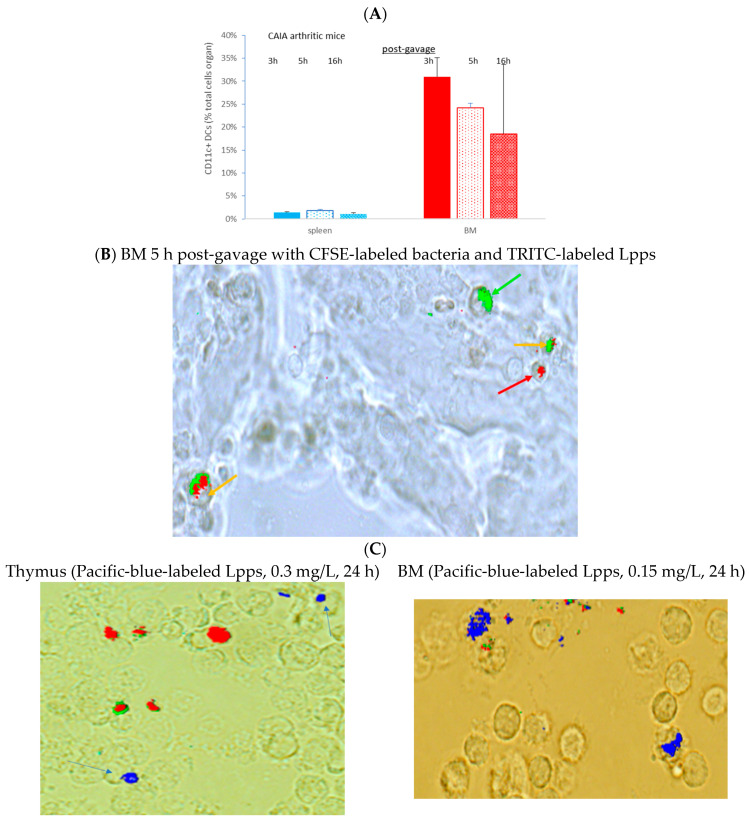
Bone marrow (BM) is the target of Lpps post-gavage. (**A**) No difference in conventional DC counts was observed in spleen and BM within 16 h post-gavage. (**B**) CFSE and TRITC signals were detected together in BM DCs from CAIA mice, indicating presence of bacteria with Lpps (orange arrow). Some cells held only green or red fluorescence, indicating the sole presence of either intestinal bacteria (green arrow) or Lpps (red arrow). (**C**) A few primary thymus and BM cells showed exclusive blue fluorescence within 24 h contact with Pacific-blue-labeled Lpps. The blue arrows point to the thymus cells capturing PB-labelled Lpps. Green and red wavelength filters were used to monitor the dying cells. (**D**) When no CFSE was added to the gavage solution, CAIA BM cells showed lower DC content. Their concentration was still higher than in BM suspensions from naïve mice (*p* < 0.0034) even force-fed with Lpps (*p* < 0.0316). Plasmacytoid cells (mPDCA) counts did not differ in BM from CAIA and naïve mice.

**Figure 5 nutrients-15-04861-f005:**
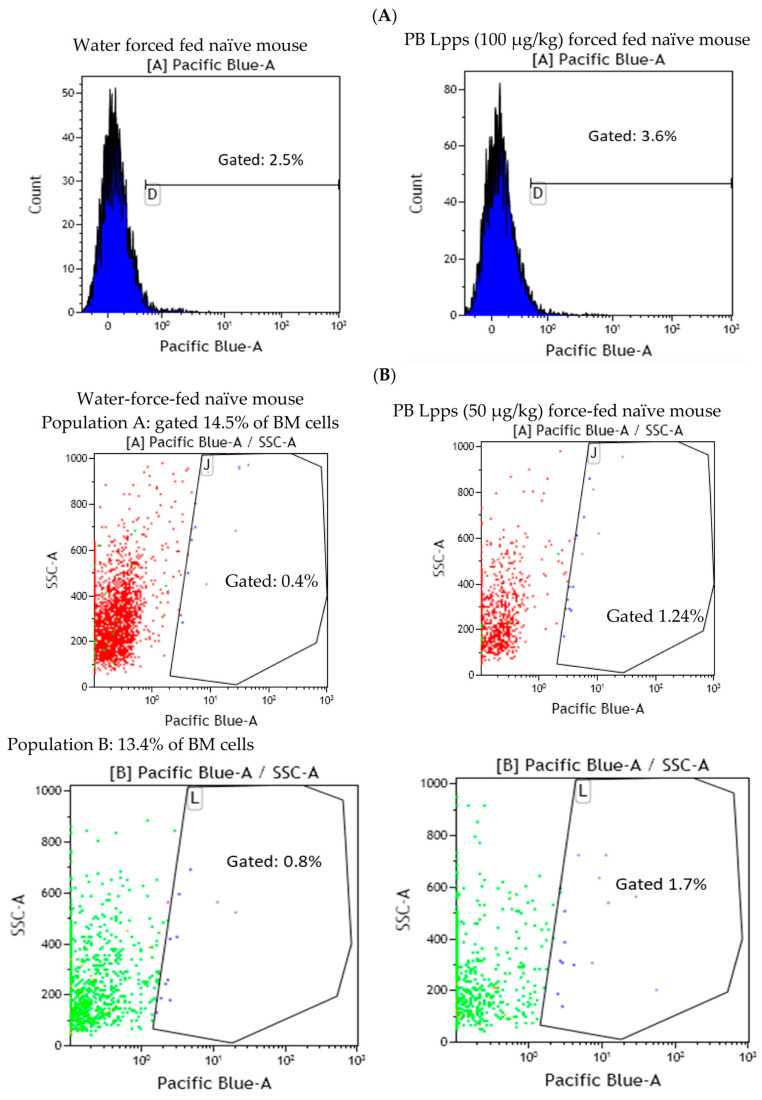
Lpps signal was primarily detected in the largest cells comprising mainly plasmacytoid dendritic cells (PDC TREM). (**A**) Force feeding naïve mice (BM cell pool from 2 mice) with 100 µg of Pacific-blue-labelled (PB) Lpps/kg led to the detection of blue signal in a slightly higher number of cells (3.6%) gated in the largest cells (A) as compared with water-force-fed control mice (2.5%). (**B**) Red cells (red dots) correspond to the largest size, and green ones (green dots) to the middle size. Pacific blue signal (PB) was detected in the two gated cells in a higher number of cells in the Lpps-force-fed mouse as compared to the water one (1.2 and 1.7% vs. 0.4 and 0.8, respectively). (**C**) Screening according to the size of cells, i.e. on the left side: the whole BM cell population on the top, the smallest BM cell population in the middle (blue dots), the largest BM cell population at the bottom (mainly red dots) indicated that the largest cells are primarily plasmacytoid cells (on the right side).

## Data Availability

The original contributions presented in the study are included in the article; further inquiries can be directed to the corresponding authors.

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
