# Peer review of "Gut–Joint Axis: Impact of Bifidobacterial Cell Wall Lipoproteins on Arthritis Development"

_nutrients, 2023, doi:10.3390/nu15234861_

Round 1

Reviewer 1 Report

Comments and Suggestions for Authors

The study investigated the protective potential of Bifidobacterium longum cell wall lipoproteins shown to modulate the intestinal microbiome and prevent osteoarthritis.

L93 was the Lpps commercial bought.

L95 I am assuming there are three treatment groups plus a no-treatment control, as well as a control for the CIA

L91 and L112 did you use both CIA and CAIA models?

L114 what source is the LPS?

L130 could you please add a brief description of the qPCR – e.g...   Followed by NGS library preparation, software, used etc.

L180 can you be more specific on what data was analysed by PCA?

L185 What p values did you deem statistically significant?

L212 what data points are p<0.05 from the graph it looks like all are correct

L232 What is the p-value of Lpps 1.0

L233 Figure 1 needs to be formatted appropriately, add all p values, properly labelled to the figure legend- and combined into one figure.

L233 figure legend should include what the plots represent e.g., box and whisker, mean and SD etc., number of replicates etc.  

L334 are both p values 0.0017.

L412 B “CFSE labelled bacteria”, is this correct

L433 second image not labelled.

L449, do you have controls for the three images, please add them in.

L510, can you add higher-resolution images?

L525 add references.

It would be useful to add the bacterial abundance/quantification data from the  2.3 Bacterial enumeration in organs

Discussion should recap your results and offer an explanation/assumptions

Comments on the Quality of English Language

The English is good, however, the methods section was somewhat confusing. 

Author Response

We thank the reviewer for the helpful comments..

1) L93 was the Lpps commercial bought.

As mentioned L95, the Lpps were prepared as described in the cited article.

We have added the following paragraph (highlighted in yellow in the text):

Briefly, bifidobacteria were grown in a broth containing lactose70 g/L, hydrolyzed milk proteins 10 g/L and cysteine hydrochloride 0.03 g/L for 48h at 37°C in anaerobic conditions. The supernatant was collected by high-speed centrifugation, concentrated by means of serial ultrafiltration. The retentate comprising molecular weight above 100kDa was then extensively dialyzed and lyophilized. Residual contamination with broth proteins was checked by subjecting the retentate to gel filtration chromatography onto Superdex 200® (Sigma–Aldrich, St. Quentin Fallavier, France).

2) L95 I am assuming there are three treatment groups plus a no-treatment control, as well as a control for the CIA.

There are 3 Lpps treatment groups (0.25, 0.5 and 1 mgLpps/L). The control group was the no-treatment group (see L111). No group was treated using drugs such Enbrel.

3) L91 and L112 did you use both CIA and CAIA models?

Yes, the two models were used. The short CAIA model was primarily dedicated to uptake analysis.

4) L114 what source is the LPS?

E.coli O111:B4 (now included in the text, line 127)

5) L130 could you please add a brief description of the qPCR – e.g...   Followed by NGS library preparation, software, used etc.

The following text was added (see highlighted text line 142)

The qPCRs were set out in 25-μl final volumes containing 5 μl of DNA template (10 ng), 12.5 μL of HOTPol Evagreen qPCRSupermix (Euromedex, Strasbourg, France) and optimized concentrations of the primers. The amplification conditions were 5-min denaturation at 95°C, followed by 40 cycles of 30 s at 95°C, 30 s for annealing at a temperature depending of primer set and 30 s at 72°C, and a final extension at 72°C for 7 min (Mastercycler realplex, Eppendorf). Melting curve analysis and migration onto 2% agarose gel (Eurogentec, Liège, Belgium) of PCR products from samples and isolated control bacteria were carried out. PCR products from isolated bacteria were purified using the Nucleospin Gel and PCR Clean Up Kit (Macherey Nagel) according to the manufacturer's instructions. NGS was performed using Illumina Miniseq (Genoscreen, Lille, France). The qPCR assays were calibrated using 0.01 fg up to 10 pg (i.e. 42 to 4.2×107 copies) of PCR-amplified target gene sequence from corresponding taxa.

6) L180 can you be more specific on what data was analysed by PCA?

Line 204: The sentence “Clustering of the data (microbiome, cells counts…) was analyzed using Principal component analysis” was replaced by: “Clustering of the data (body weight, spleen weight, splenocytes/mg spleen, bacteria in any intestinal location, splenic CD11c+ and CD4+ cells counts, number of Peyer’s patches) was analyzed using Principal component analysis”

7) L185 What p values did you deem statistically significant?

We added the missing sentence: P values less than 0.05 were considered significant. And we deleted the confusing part of the sentence related to post-hoc tests.

8) L212 what data points are p<0.05 from the graph it looks like all are correct.

We apologize for the lack of clarity. We used mixed model for repeated measures to test the hypothesis that the AI curves were (or not) significantly different. We did not repeat the statistical analysis at each observational point.

Accordingly, the sentence “As showed on Figure 1A, Lpps dosing at 0.5 but not 1 mg/L alleviated CIA progression (p<0.05)” was changed Line 216 as follows:” Statistical analysis of the AI curves using the mixed model  for repeated measures (Figure 1A), indicated that  Lpps dosing at 0.5 but not 1 mg/L alleviated CIA progression (p<0.05)”

9) L232 What is the p-value of Lpps 1.0:

p=0.2

10) L233 Figure 1 needs to be formatted appropriately, add all p values, properly labelled to the figure legend- and combined into one figure.

11) L233 figure legend should include what the plots represent e.g., box and whisker, mean and SD etc., number of replicates etc. 

We hope the new version of Figure 1is clearer and more appropriate (see L 247)

12) L334 are both p values 0.0017.

We apologized for the lack of clarity. There is only a p value pointing out the group difference.

To clarify, the sentence in the legend (line 342) was changed as follows:

There is a group effect, i.e. splenocytes from RA mice exhibited fewer CD11c+ DCs than CIA mice (p=0.0017). The dosing did not significantly affect CD11c+ DCs counts

In the figure, a horizontal bracket was added and the p value put above.

13) L412 B “CFSE labelled bacteria”, is this correct

CFSE passes across the membrane of cells in its close neighboring. Therefore, bacteria and primarily epithelial cells from the intestine are becoming fluorescent when CFSE is given orally. Presence of green fluorescence in cells from organs distant from the intestine indicated that bacteria crossed the intestinal barrier and were captured by cells located in distant organs.

To clarify we tried to improve the figure and changed the sentence (line 406) as follows:

14) B: : CFSE and TRITC signals were detected together in BM DCs from CAIA mice indicating presence of bacteria with Lpps (orange arrow). Some cells held only green or red fluorescence indicating the sole presence of either intestinal bacteria (green arrow) or Lpps (red arrow).

15) L433 second image not labelled.

We modified the size of the photographs to provide a better view on the cells with Pacific blue and on the labeling.

16) L449, do you have controls for the three images, please add them in.

When mice were forced fed with the sole TRITC, no signal was seen in cell from the targeted organs.

17) L510, can you add higher-resolution images?

Unfortunately, the images sent by the technical platform cannot be further improved

18) L525 add references.

Since we have not already published these results, we added “unpublished personal data”

19) It would be useful to add the bacterial abundance/quantification data from the 2.3 Bacterial enumeration in organs.

The few bacteria showing quantitative difference were presented in the text and in figure 1. We add two tables (Table S1 referred to line 236, Table S2 line 2) as supplementary results to give an insight on the most frequent species.

20) Discussion should recap your results and offer an explanation/assumptions.

We added in the first paragraph the main observations, e.g. the limited involvement of the intestinal bacteria already shown to have a prominent role in osteoarthritis, and in contrast the key point of transport to the inflamed sites, infiltrated with high DCs counts. We then focused onto the dosing aspect and tuned our conclusion about the intestinal bacteria as already made in the previous version

Reviewer 2 Report

Comments and Suggestions for Authors

In the manuscript by Piva et al., they showed that  the administration of 0.5 mg of Lpps per liter (0.5 mg Lpps/L) and 1 mg of Lpps per liter (1 mg Lpps/L) demonstrated a significant reduction in rheumatoid arthritis (RA) symptoms in CIA DBA/100aHsd mice. These mice developed arthritis either through collagen-induced arthritis (CIA) or by injecting anti-collagen antibodies (CAIA)

In line 45 I think it is a typo, the size of some letters in this line is bigger then normal.

In Figure 1B Lpps missing 0,5 mg/L Lpps

Why in line 11 you are mentioning 0,25 concentration you don’t show this concertation diagrams

Why you used collagen and collagen antibody is the mechanism the same?

The expression control in the diagrams is confusing in Figure 1. Expression control is usually used for the samples or mice that are not treated with anything, while here we have a treatment in all mice and you are analyzing if addition of Lpps have a beneficial impact. Would be better to show if like CIA, CIA + 0,5 mg/L Lpps, CIA+1 mg/L Lpps, would be even more reliable if you would really have a control plain Black/6 mice +/- Lpps. Labelling of controls is confusing also in Figure 3 once is control then another time CIA control. What is now plain control and what CIA control

PC graph in my opinion are not informative enough, this data visualization could be better done.

In Figure 3 there is no indication what means C1, C4, C5, C8. Labeling of groups is poor through the whole paper.In figure 3A there are huge variation in the groups with 4 replicate and between groups.

Did you try to analyze the presence of pacific blue-Lpps in small intestine (IEL fraction and LP fraction) or colon in IELfr or LPfr?

Did you try to analyze DC also in the small intestine (IELfr and LPfr) or colon(IELfr and LPfr)?

Author Response

We would like to thank the reviewer for the valuable comments.

In the manuscript by Piva et al., they showed that  the administration of 0.5 mg of Lpps per liter (0.5 mg Lpps/L) and 1 mg of Lpps per liter (1 mg Lpps/L) demonstrated a significant reduction in rheumatoid arthritis (RA) symptoms in CIA DBA/100aHsd mice. These mice developed arthritis either through collagen-induced arthritis (CIA) or by injecting anti-collagen antibodies (CAIA)

1) In line 45 I think it is a typo, the size of some letters in this line is bigger then normal.

We apologize for this inconvenience. Correction is done.

2) In Figure 1B Lpps missing 0,5 mg/L Lpps

The missing Lpps is now present.

3) Why in line 11 you are mentioning 0,25 concentration you don’t show this concertation diagrams

There were 3 treatment groups in addition to the control group corresponding the drinking water group. Since the figure is quite complex, we kept only the 1mg Lpps/L as the unprotective dosage.

4) Why you used collagen and collagen antibody is the mechanism the same?

Both models induced arthritis through anti-collagen antibodies. The inducer of anti-collagen antibodies is collagen (plus Freund’s adjuvant) in the CIA model. Administration of the anti-collagen antibodies made the model shorter.

5) The expression control in the diagrams is confusing in Figure 1. Expression control is usually used for the samples or mice that are not treated with anything, while here we have a treatment in all mice and you are analyzing if addition of Lpps have a beneficial impact. Would be better to show if like CIA, CIA + 0,5 mg/L Lpps, CIA+1 mg/L Lpps, would be even more reliable if you would really have a control plain Black/6 mice +/- Lpps. Labelling of controls is confusing also in Figure 3 once is control then another time CIA control. What is now plain control and what CIA control

We apologize for not being clear enough. We mention in the Material and Methods section line 111: “Control group (n= 8 per assay) received sterile water instead of Lpps solution”

We put CIA with the word “control” when the comparison with RA mice was carried out.

6) PC graph in my opinion are not informative enough, this data visualization could be better done.

We tried to improve the reading by putting PCA graphs on the same line (Figure 1, line 250). It provides at a glance the information about the factors influencing the arthritis index (only significant contributions were colored).

7) In Figure 3 there is no indication what means C1, C4, C5, C8. Labeling of groups is poor through the whole paper.In figure 3A there are huge variation in the groups with 4 replicate and between groups.

We apologize for the lack of information. The complementary explanation is now provided in the legend of Figure 3 (number of each mouse, CIA vs RA mice...) (see line338). The statistical analysis is based on mixed model for repeated measures thus minimizing the huge difference in cell response to LPS elicitation.

8) Did you try to analyze the presence of pacific blue-Lpps in small intestine (IEL fraction and LP fraction) or colon in IELfr or LPfr?

We did not attempt to detect Pacific blue in the small intestine for there is contaminating fluorescence from the diet (even though we used a specific diet). We used several fluorochrome without evidencing a fluorescent detection limit compatible with the Lpps dosage.

9) Did you try to analyze DC also in the small intestine (IELfr and LPfr) or colon(IELfr and LPfr)?

No for the same reason

Reviewer 3 Report

Comments and Suggestions for Authors

This study demonstrated that cell wall lipoprotein of Bifidobacterium longum had beneficial effects in a arthritis development mouse model. The results are interesting, however, there is no clear description of study purpose. And preparation and English of the manuscript are a little poor.

Major revisions

1.        Line 73-80: The author should clearly state the purpose of the study.

2.        In this manuscript, there are two descriptions for the dose of Lpps: 0.25, 0.5 and 1 mg/L, and 25, 50 and 100 μg Lpps/kg. The author should unify as the latter description.

3.        Line 234-235: The author described “arthritis index was significantly lower in 0.5mg Lpps/L (p<0.05)”. The author should revise this text to make it clear at what point or period and what kind of analysis resulted in a significant difference between the control group and 0.5 mg Lpps/L.

4.        Figure 3B: The author should revise the bar positions in the Figure to show which groups have between-group differences. In this case, CIA cont vs RA cont and CIA Lpps 1 vs RA Lpps 1 should be compared.

5.        Line 373-374: Figure 3C shows that the recruitment of not only CD4+ T and CD4+CD25+ Treg cells but also CD11c cells were reduced by Lpps intake in female but not male mice. It should be revised.

6.        I think too that the clarification of the level of lipoproteins in human gut is important. However, there are various species of endogenous bifidobacteria and different abundances of them in colon even in healthy people. Different species of bifidobacteria may have different composition of lipoproteins, which have different activity. Is the developed affinity column specific to capture Bifidobacterium longum Lpps or active Lpps? This point should be discussed more.

7.        Line 613-914: What can be considered as regulating the lipoprotein release at the bacterial level? Is it autolysis or a special secretion mechanism? The author should describe the possibilities.

Minor revisions

1.        Line 51: an rident model a rodent model

2.        Figure 4D: p<0.0034? Is it p<0.034 as described in Line 458 or p=0.034?

Comments on the Quality of English Language

The preparation and English of the manuscript are a little poor.

Author Response

We would like to thang the reviewer for the helpful comments.

This study demonstrated that cell wall lipoprotein of Bifidobacterium longum had beneficial effects in a arthritis development mouse model. The results are interesting, however, there is no clear description of study purpose. And preparation and English of the manuscript are a little poor.

Major revisions

  1. Line 73-80: The author should clearly state the purpose of the study.

We changed the last paragraph as follows:

The purpose of the present study was to evaluate the protection induced by Lpps intake in an arthritis background and decipher whether the gut bacteria were involved in the beneficial effect and/or Lpps were able to reach the inflamed joint area. Free lipoproteins from B.longum were administrated in collagen-induced (CIA) and anti-collagen antibody induced arthritis (CAIA) DBA/1OOaHsd mouse models and the arthritis index was scored. Selected bacteria shown to be targeted by Lpps in human and animal models were quantified along the intestine at the end of the surveys. To further investigate the possible impact of the microbiome, we included gnotobiotic mice associated with RA patient’s microbiota as a surrogate of arthritis initiation and we compared IL-10 responses to lipopolysaccharides elicitation of CIA and RA splenocytes. In addition, bone marrow was targeted as a compartment close to the destroyed joint harboring conventional and plasmacytoid dendritic cells (DCs) able to capture Lpps following a possible passage across the intestinal barrier. Uptake of Lpps was studied in vitro and in vivo by administrating labelled BC to CAIA and healthy mice.

  1. In this manuscript, there are two descriptions for the dose of Lpps: 0.25, 0.5 and 1 mg/L, and 25, 50 and 100 μg Lpps/kg. The author should unify as the latter description.

We would like to keep with the two versions because the concentration is more accurate than the estimation of dosage for the long-term survey (CIA). We did not keep with gavage for the chronic stress abrogated the protection. The short term assay (CAIA) was used for introducing the notion of dosing of Lpps per kg. We recall the concentration used in the title of the figure 2 (see line 311)

  1. Line 234-235: The author described “arthritis index was significantly lower in 0.5mg Lpps/L (p<0.05)”. The author should revise this text to make it clear at what point or period and what kind of analysis resulted in a significant difference between the control group and 0.5 mg Lpps/L.

We apologize for the lack of clarity. We used mixed model for repeated measures to test the hypothesis that the AI curves were (or not) significantly different. We did not repeat the statistical analysis at each observational point.

Changes were made as follows:

In the text line 216:

Statistical analysis of the AI curves using the mixed model for repeated measures (Figure 1A) indicated that Lpps dosing at 0.5 but not 1 mg/L alleviated CIA progression (p<0.05).

And in the legend:

A:  arthritis index curve of 0.5mg Lpps/L group was significantly different from that of the control curve (vertical bracket, p<0.05, mixed model for repeated measures).

We did change also the presentation of Figure 1A to clarify the type of statistical analysis used.

  1. Figure 3B: The author should revise the bar positions in the Figure to show which groups have between-group differences. In this case, CIA cont vs RA cont and CIA Lpps 1 vs RA Lpps 1 should be compared.

We apologized for the lack of clarity. There is only a p value pointing out the group difference. To clarify, the sentence in the legend was changed as follows:

There is a group effect, i.e. splenocytes from RA mice exhibited fewer CD11c+ DCs than CIA mice (p=0.0017). The dosing did not significantly affect CD11c+ DCs counts

In the figure, a horizontal bracket was added and the p value put above.

  1. Line 373-374: Figure 3C shows that the recruitment of not only CD4+ T and CD4+CD25+ Treg cells but also CD11c cells were reduced by Lpps intake in female but not male mice. It should be revised.

Actually, there is no significant CD11c DCs reduction following Lpps in female mice. The observed difference applied to the comparison between Lpps treated male and female mice.

We added the following sentence in the legend line 346.

In contrast, following treatment, the female spleen contained less CD11c+ DCs than male ones (p=0.029)

  1. I think too that the clarification of the level of lipoproteins in human gut is important. However, there are various species of endogenous bifidobacteria and different abundances of them in colon even in healthy people. Different species of bifidobacteria may have different composition of lipoproteins, which have different activity. Is the developed affinity column specific to capture Bifidobacterium longum Lpps or active Lpps? This point should be discussed more.

CV B4 can capture at least B.breve as well as B.longum Lpps. It is more likely that the columns are specific to active Lpps, but the range of Lpps recognized by the viral particles remains to be ascertained.

We added the comment onto at least B.breve and B.longum Lpps (line 507). 

  1. Line 613-914: What can be considered as regulating the lipoprotein release at the bacterial level? Is it autolysis or a special secretion mechanism? The author should describe the possibilities.

we gave some further information of the state of the art at least at the synthesis level (Line 517). However, there is questions about the maturation of Lpps. Thus speculation about release was made short, not even about a possible autolysis which is not obvious during fermentation.

Minor revisions

  1. Line 51: an rident model → a rodent model

Correction is made

  1. Figure 4D: p<0.0034? Is it p<0.034 as described in Line 458 or p=0.034?

Thank you for pointing the discrepancy; the correct value is p<0.0034 (modification is made in the text)

Reviewer 4 Report

Comments and Suggestions for Authors

This reviewed manuscript is devoted to studying the influence of individual microbiota components on the progression of rheumatoid arthritis. The topic is undoubtedly relevant and important. In general, the manuscript contains comments only on the methodological part of the work.

The authors should briefly describe the techniques to which they refer, as described previously. First, in some cases, in the articles they refer to, you can also find a similar reference link - as described previously, so it is difficult to present the methodology as a single whole. Second, as with the methodology for "Bacterial enumeration in organs", the question arises regarding the correctness of the presentation of the results obtained. Thus, in the article referred to by the authors, 18 bacterial indicators were analyzed. The authors themselves present results for only seven of them. What is the situation with the other indicators? Were they not analyzed? Or they are not indicative - then why? Neither in the text of the methodological part nor in the main text of the results ​​​​​​​the authors provide a complete list of bacterial indicators that were analyzed.

Author Response

We would like to thank the reviewer for the comments.

This reviewed manuscript is devoted to studying the influence of individual microbiota components on the progression of rheumatoid arthritis. The topic is undoubtedly relevant and important. In general, the manuscript contains comments only on the methodological part of the work.

The authors should briefly describe the techniques to which they refer, as described previously. First, in some cases, in the articles they refer to, you can also find a similar reference link - as described previously, so it is difficult to present the methodology as a single whole. Second, as with the methodology for "Bacterial enumeration in organs", the question arises regarding the correctness of the presentation of the results obtained. Thus, in the article referred to by the authors, 18 bacterial indicators were analyzed. The authors themselves present results for only seven of them. What is the situation with the other indicators? Were they not analyzed? Or they are not indicative - then why? Neither in the text of the methodological part nor in the main text of the results ​​​​​​​the authors provide a complete list of bacterial indicators that were analyzed.

We apologize for not being clear enough. We added several descriptions of the techniques used (see paragraphs from line 94 and from line 142). We also added two supplementary tables with the results for the species we were able to detect. As you will see in the tables, our attention focused mainly onto bacteria implicated in the response to Lpps in the osteoarthritis model, i.e. Parabacteroides goldsteinii, Akkermansia mucinophila, Mucispirillum schaedlerii, the three lactobacilli and Eubacterium plexicaudatum. A few other ones (primarily the Schaedler species) were included since they were found using culturomics. Signals were not always detected concerning Shaedler bacteria likely because the osteoarthritis microbiome was oversimplified as compared with DBA1 models. The inflammation is also a factor probably affecting the microbiome.

Round 2

Reviewer 1 Report

Comments and Suggestions for Authors

I am happy with the update mansucript

Reviewer 2 Report

Comments and Suggestions for Authors

I already put the comment in the previous revision. For this version i have no comments.

Reviewer 3 Report

Comments and Suggestions for Authors

The manuscript has been well revised.